

**Climate effects on the vitality of boreal forests at the treeline in different ecozones of**
**Mongolia**
Michael Klinge[1], Choimaa Dulamsuren[2], Stefan Erasmi[1], Dirk Nikolaus Karger[3], Markus
Hauck[2]
[1]  Institute of Geography, University of Goettingen
Goldschmidtstr. 5, D-37077 Goettingen, Germany
[2]  Albrecht-von-Haller Institute for Plant Sciences
Plant Ecology and Ecosystems Research, University of Goettingen
Untere Karspuele 2, D-37073 Goettingen, Germany
[3]  Swiss Federal Research Institute WSL, Züricherstrasse 111, 8903 Birmensdorf, Switzerland
**Abstract**
In northern Mongolia, at the southern boundary of the Siberian boreal forest belt, the distribution of
steppe and forest is generally linked to climate and topography, making this region highly sensible to
climate change. Detailed investigations on the limiting parameters of forest and steppe occurrence in
different ecozones provide necessary information for environmental modelling and scenarios of
potential landscape change. In this study, remote sensing data and gridded climate data were
analyzed in order to identify distribution patterns of forest and steppe in Mongolia and to detect
driving ecological factors of forest occurrence and vulnerability against environmental change. With
respect to anomalies in extreme years we integrated the climate and land cover data of a 15 year
period from 1999-2013. Forest distribution and vegetation vitality derived from the normalized
differentiated vegetation index (NDVI) were investigated for the three ecozones with boreal forest
present in Mongolia (taiga, subtaiga, and forest-steppe). In addition to the entire ecozone areas, the
analysis focused on different subunits of forest and non-forested areas at the upper and lower
treeline, which represent ecological borderlines of site conditions.
The total cover of boreal forest in Mongolia was estimated at 73,818 km$^2$. The upper treeline
generally increases from 1,800 m above sea level (a.s.l.) in the Northeast to 2,700 m a.s.l. in the
South. The lower treeline locally emerges at 1,000 m a.s.l. in the northern taiga and is rising
southward to 2,500 m a.s.l. The latitudinal trend of both treelines turns into a longitudinal trend in
the east of the mountains ranges due to more aridity caused by rain-shadow effects. Less vital trees
were identified by NDVI at both, the upper and lower treeline in relation to the respective ecozone.
The mean growing season temperature (MGST) of 7.9-8.9 °C and a minimum of 6 °C was found to be
a limiting parameter at the upper treeline but negligible for the lower treeline and the total
ecozones. The minimum of the mean annual precipitation (MAP) of 230-290 mm y$^{-1}$ is an important
limiting factor at the lower treeline but at the upper treeline in the forest-steppe ecotone, too. In
general, NDVI and MAP are lower in grassland, and MGST is higher compared to the forests in the
same ecozone. An exception occurs at the upper treeline of the subtaiga and taiga, where the alpine
vegetation is represented by meadow mixed with shrubs. Comparing the NDVI with climate data
shows that increasing precipitation and higher temperatures generally lead to higher greenness in all
ecological subunits. While the MGST is positively correlated with the MAP of the total ecozones of
the forest-steppe, this correlation turns negative in the taiga ecozone. The subtaiga represents an
ecological transition zone of approximately 300 mm y$^{-1}$ precipitation, which occurs independently
from the MGST. Nevertheless, higher temperatures lead to higher vegetation vitality in terms of
NDVI values.



Climate change leads to a spatial relocation of tree communities, treelines and ecozones, thus an
interpretation of future tree vitality and biomass trends directly from the recent relationships
between NDVI and climate parameters is difficult. While climate plays a major role for vegetation
and treeline distribution in Mongolia, the disappearing permafrost needs to be accounted for as a
limiting factor for tree growth when modeling future trends of climate warming and human forest
disturbance.

## 1. Introduction

Due to the highly continental environment in northern Central Asia, Mongolia is subjected to dry and
cool climate conditions. The landscape and vegetation development is highly sensitive to changes in
temperature and/or precipitation. However, this is not a uniform phenomenon throughout the entire
region. The intensity and impact of climate parameters on vegetation is strongly varying in space and
time caused by different factors like topography, latitude and air circulation. Corresponding to the
change of the climatic conditions from cold semi humid in the north to arid in the south a latitudinal
zonation of the vegetation occurs, which is modified by an altitudinal zonation in the mountainous
landscape (Hilbig, 1995). From north to south, these vegetation belts include taiga, subtaiga, forest-
steppe, steppe, and the Gobi desert. Taiga, subtaiga, and forests in the forest-steppe ecotone
represent the southern edge of the Eurosiberian boreal forest, whereas the steppes are part of the
Mongolian-Chinese steppe region. The distribution of the different vegetation belts, ecozones, and
treelines is controlled by air temperature, evapotranspiration, and precipitation (Walter and Breckle,
1994). Moisture conditions are regarded to be a main limiting factor for the distribution of the desert
and steppes ecozones as well as for the lower boundary of mountain forests at the transition to
drylands. In contrast, thermal conditions control the upper treeline and the alpine ecozone (Körner,
2012; Klinge et al., 2003, 2015; Paulsen and Körner, 2014). Both, the upper and the lower treeline of
Mongolia's boreal forests represent an obvious visual boundary between biomes of highly different
ecological requirements, though their actual state can be strongly influenced by human impact
(Klinge et al., 2015). Trees grow and exist for several decades or centuries and establish an
autochthonous microclimate below the canopy, thus forests are representing mean climatic
conditions of a longer period. In contrast, the vitality of annual or perennial grasses and herbs of the
steppes and meadows respond to inter-annual variation in climate conditions and the vegetation
density represents small-scale periods (Bat-Oyun et al., 2016).
Mean air temperature during the growing season (MGST) is more relevant for describing the thermal
environment at the upper forest line than mean annual air temperature, because temperatures from
the non-growing cold season only play a minor role in tree growth (Jobbágy and Jackson, 2000;
Körner, 2012; Körner and Paulsen, 2004). Based on worldwide empirical data Körner and Paulsen
(2004) stated that the minimum MGST of 5.5 to 7.5 °C and the mean temperature of 6.4 °C during a
period of daily temperatures >0.9 °C in a minimum growing season of 94 days (Paulsen and Körner,
2014) are better definitions for the upper treeline in a global context than the commonly used
warmest month isotherm of 10 °C (Walter and Breckle, 1994). A lower treeline occurs in the semi-
arid region of Central Asia between relatively humid mountain regions and arid basins. The forest
distribution is generally limited by annual precipitation, which has its minimum between 300 and 200
mm y$^{-1}$ (Dulamsuren et al., 2010a; Holdridge, 1947; Miehe et al., 2003; Walter and Breckle, 1994).
Dulamsuren et al., (2010a) proved an annual precipitation between 230 and 400 mm for larch trees
(*Larix sibirica*) at the lower forest boundary in northern and central Mongolia. However, additional
soil water supply from upslope and from melting permafrost ice supports tree growth at lower
elevations where rainfall is insufficient. Furthermore, drought periods can be temporarily bridged by
the soil ice reservoir. This explains why Dulamsuren et al. (2014) found coniferous forests in regions
with an annual precipitation of around 120 mm in the Altai Mountains in western Mongolia.





Dulamsuren and Hauck (2008) and Dulamsuren et al. (2010a, 2010b) investigated the ecological
conditions in the forest-steppe ecotone of Mongolia, where steppe and forest alternate in short
distances. In the forest-steppe, the spatial distribution of vegetation is highly correlated with relief
parameters (Hais et al., 2016; Klinge et al., 2015). Less solar radiation input causes lower
temperatures and reduces the evapotranspiration pressure on north-facing slopes, leading to higher
humidity, higher soil moisture, and more widespread permafrost. The higher water availability
supports the growth of trees, which is Siberian larch (*Larix sibirica*) on most of Mongolia's forested
area (Dashtseren et al., 2014). On south-facing slopes more solar radiation input produces
hydrological conditions which are too dry for the establishment of forests and thus favor steppe
vegetation (Bayartaa et al., 2007).
With respect to global climate change, the question of potential shifts in growth conditions arises.
Vegetation indices like the most commonly applied NDVI (Normalized differentiated vegetation
index), which are derived from multispectral satellite images (Landsat, MODIS, Spot VGT) provide
information about the "greenness" and vitality of the vegetation cover. The various investigations on
recent trends of climate and NDVI, which exist for the region of Mongolia state partially diverging
results (Dashkhuu et al., 2015; Eckert et al., 2015; Miao et al., 2015; Poulter et al., 2013; Vandandorj
et al., 2015). Instrumental climate data from weather stations in Mongolia are often discontinuous
and time series of climate measurements are not available from mountain areas since climate
stations are located near settlements in the basins. Thus, representative climate parameters must be
modelled by different regionalization processes (Böhner, 2006). Various gridded datasets of re-
analyzed climate parameters with different spatial and temporal resolution, which are mainly used
for climate trend analysis, exist: e.g. CRU-TS (Harris et al., 2014), ERA-interim (Dee et al., 2011),
CHELSA (Karger et al., 2016). While the quality, origin, and resolution of climate records constitute
one uncertainty factor, the results and interpretations about the correlations between climate and
NDVI trends occasionally suffer from disregarding the specific bio-ecological restrictions of the
different vegetation zones.
Batima et al. (2005) analyzed climate station data and observed an increasing mean annual air
temperature (MAAT) of 1.66 °C for Mongolia between 1940 and 2001. Eckert et al. (2015) stated that
temperatures have not varied much since the year 2000. Dulamsuren et al. (2014) found a trend to
warmer temperature extremes starting around 2000. Sharkhuu et al. (2007) and Sharkhuu (2003)
executed measurements on permafrost distribution and active layer development in Mongolia for
more than 30 years. They found a general trend of permafrost degradation, which is additionally
accelerating since the 1990s. This is due to climate warming, but reinforced by a loss of vegetation
due to livestock grazing in some steppe areas and tree cutting in the forests. Permafrost degradation
is more intense in the Khuvsgul area than in the Khentei and Khangai Mountains.
The trends of precipitation in Mongolia are not spatially uniform. However, the observed trend can
strongly depend on the specific period used for climate analysis (Erasmi et al., 2014; Giese et al.,
2007).  This can explain the different results between Batima et al. (2005) and Eckert et al. (2015),
who analyzed the climate development in Mongolia at different time spans in the period, which was
regarded in this research. While there was a positive trend in the annual precipitation found in the
forest regions of northern and central Mongolia during the period from 2001 to 2011, it has been
negative in the previous period between 1970 and 2001. In the driest regions of western and
southern Mongolia however, no specific trends occurred at all. Based on tree-ring data, Dulamsuren
et al. (2010b) documented increasing drought stress in larch trees in the Khentei Mountains, which
they attributed to increasing aridity by rising summer temperatures and decreasing summer
precipitation during the last 50 years. Although trees at the outer boundary of the forest stands
might be better adapted to drought stress, obvious margins of dead trees surrounding the forest



islands are recently found at many places of the forest-steppe. For the period from 1980 until 2005,
Bayartaa et al. (2007) reported a strong increase in burnt forest area in Mongolia starting in 1996,
which was due to very dry winter and spring seasons but may also be combined to weakened
governmental management during the period of political transition. A general tendency of
decreasing lake levels during the last decades in two great lakes of interior drainage in the Gobi with
an catchment area south of Khangai Mountains was observed by Szumińska (2016). This lake level
decline was associated with trends for reduced precipitation and increased evapotranspiration
resulting from rising temperatures.
Eckert et al. (2015) analyzed the general trend of NDVI in Mongolia during the period between 2001
and 2011 using the MODIS NDVI dataset and found mostly positive trends in northern and eastern
Mongolia, stable conditions in southern Mongolia, and large areas of negative trends in the northern
Mongolian Altai and in the east of the Khangai Mountains. Based on the same dataset and a similar
period from 2000 to 2012, Vandandorj et al. (2015) analyzed the seasonal variation of NDVI for
individual vegetation zones. High variations in NDVI occur particularly in the steppe regions where
the vitality and density of grassland is closely related to the amount of annual precipitation due to
low stomatal control of transpiration by the grassland vegetation. Low variations in NDVI occur in
forested regions, since trees exert a much stricter stomatal control of transpiration than herbs and
grasses, and in the sparsely vegetated desert regions. Poulter et al. (2013) investigated the influence
of recent climate trends on the forests in Inner Asia by the temporal distribution of a greening value
using specific vegetation indices from remote sensing data and environmental datasets. They found a
trend to earlier greening induced by increasing spring temperatures and earlier browning associated
with decreasing summer precipitation. Based on these relationships they projected better future
forest conditions for Mongolia until 2100. In opposite of these findings, Bayartaa et al. (2007)
reported that climate scenarios would indicate a significant decrease in forest area and its total
biomass for Mongolia until the middle of the 21$^{st}$ Century, which is in accordance with the recent
trends from dendrochronological data from Mongolia (Dulamsuren et al., 2010a, 2010b, 2014;
Khansaritoreh et al., 2017). Lu et al. (2014) investigated the applicability of different remote sensing-
based biomass estimation approaches. They found the biomass estimation method via NDVI to be
sufficient in low density forests. Dulamsuren et al. (2016) showed that the NDVI well usable to
estimate the tree biomass for Mongolian forests. The best fit of linear regression was found between
biomass and the mean NDVI of April for the period 1999-2013. This shows that in addition to the
vegetation vitality the NDVI is a valuable indicator for tree biomass in open forest stands.
With regard to the diverse and in parts contradicting observations on climate and vegetation status,
interdependencies and recent trends in Mongolia that are reported here, this study investigates the
present distribution of forest areas and its relation to climate and topography based on high
resolution satellite and gridded climate data. In addition to existing studies, here, the impact of
climate and changes in climate parameters is studied at different spatial levels related to the
zonation of ecozones. The following hypotheses were tested:
• Every ecozone has its own climatic restricted environment. The statistical correlations between
NDVI and climate condition in different forest types and at the corresponding treelines reflect
the specific ecological relationships and limitations.
• There are different trends of climate-induced vitality change detectable for the different
ecozones and especially for the treelines as an indicator for extreme ecological site conditions.
• Forests and grasslands of the same ecozone show different trends and relations to climate and
NDVI.



## 2. The Study Area

Mongolia is situated in northern Central Asia in the transition zone between the Siberian taiga in the north and the Gobi desert in the south (Fig. 1). Spatially Mongolia extends from 87°45'E to 119°56'E and from 41°34'N to 52°09'N and covers a total area of 1,562,950 km$^2$. Wide basins of interior drainage are spread on elevations between 900 and 1500 m a.s.l. with the lowest areas below 720 m a.s.l. There are five principal mountain systems in Mongolia: The Mongolian Altai (MA) in the west (highest peak is Tavan Bogd, 4374 m a.s.l.), the Gobi Altai in the south (Ikh Bogd, 3957 m a.s.l.), the Khangai Mountains (KaM) in the center (Otgon Tenger, 3964 m a.s.l.), the Khentei Mountains (KeM) in the northeast (Asralt Kharj khan, 2799 m a.s.l.), and the Khuvsgul region in the eastern Sayan Mountains (Munkh Saridag, 3460 m a.s.l.). The mountain tops are shaped by pronounced flat surfaces at elevations between 2500 and 3500 m a.s.l. (Academy of Sciences of Mongolia and Academy of Sciences of USSR, 1990; Murzaev, 1954)

The climate of Mongolia is characterized by high continental semi humid, semiarid, and arid conditions. In wintertime, the Siberian high pressure cell produces cold and dry weather with few snowfall and mean temperatures between -15 and -30 °C (Barthel, 1983; Klinge, 2001). The main rainfall occurs from June to August during the short summer and is induced by westerlies and cyclone precipitation, with the dry season starting again in autumn. The mean summer temperatures range between 10 and 27 °C. Mean annual precipitation is lower than 50 mm in the interior basins, around 125 mm in the southern desert and up to 350 mm in the northern steppes, whereas it increases to more than 500 mm in the high mountains. There is a large annual variation in precipitation amount and period, which strongly controls the annual density of the steppe vegetation cover (Bat-Oyun et al., 2016).

According to the climatic conditions, the vegetation zones occur in a latitudinal and altitudinal order (Hilbig, 1995). Dark mountain taiga with coniferous trees (*Pinus sibirica, Picea* obovata, *Abies sibirica*) occurs as closed forests in northern Mongolia and selective in the upper KaM in central Mongolia (Dulamsuren, 2004). The subtaiga ecozone with needle and deciduous broadleaf forests (*Larix sibirica, Pinus sylvestris, Betula platyphylla*) represents a type of light taiga beneath and surrounding the mountain taiga. In northern Mongolia, the forest often extends into the valley bottoms and open grassland is restricted to intra-mountainous basins. The vegetation in central Mongolia consists of steppe grasslands in the basins and forest-steppe in the mountain area. Small areas of grassland have been converted into croplands. In this forest boundary ecotone of semiarid climate conditions, the relief controls the vegetation patterns. While the deciduous conifer forests consisting of *Larix sibirica* are primarily limited to north-facing slopes, the southern slopes are covered by steppe vegetation (Treter, 1996). The southern part of Mongolia consists of desert steppe and sparse desert vegetation. Sand dunes, as well as playas and takirs, which consist of salty and clayey sediments remaining from evaporated water in episodically existing lakes in basins of interior drainage, are widely distributed. In the high mountains, dense alpine meadow vegetation occurs between forest-steppe and the periglacial zone of frost debris. The main perennial rivers are accompanied by floodplain meadows and alluvial forests (Hilbig, 1995).

Missing forestry management and extensive forest use by tree cutting and wood pasture led to forest degradation and local deforestation in many regions of Mongolia during the last decades (Tsogtbaatar, 2004). In addition, hazardous forest fires destroyed large forest areas (Bayartaa et al., 2007; Goldammer, 2002, 2007; Hansen et al., 2013). Although it is supposed that most of the recent forest fires in Mongolia were primarily set by humans, there has to be an additional ecological exposure to fire susceptibility (Dorjsuren, 2009).



## 3. Methods

An overview of the complete analysis process is illustrated in figure 2, while the single steps are
described in detail below. The spatial resolution of the various basic data sets is presented in figure 3.
The forested area was mapped for Mongolia and its surroundings using a maximum likelihood
supervised classification of 50 Landsat 8 satellite images (spatial resolution 30 m). Images of the
years 2013 and 2014 were used as a baseline, and, in areas of low quality or high cloud coverage,
were supplemented by Landsat 5 images from 2009 to 2011 (spatial resolution 30 m).

The elevation of the actual treeline was calculated by selected points from a digital elevation model
(DEM) of SRTM-data (spatial resolution 90 m). Points representing the treelines were established by
a kernel-model which evaluates for every pixel covered by forest if (1) it lies on a slope of more than
2°, (2) there is any forested area in the surroundings in a higher or lower position, and (3) there is any
woodless area representing the existence of the next vegetation zone beyond the potential forest
boundary to exclude relief related distribution limits. The specific search parameters for the upper
and lower treeline are given in figure 2. Körner (2012) proposes a minimum vertical range from the
upper treeline (UT) to the summit to prevent the summit effect on tree development and to receive
a true climatic treeline value. Due to extensive planation surfaces in the investigation area of KaM,
flat mountaintops in the alpine zone widespread occur. During the analysis process, it was necessary
to reduce the minimum distance between the upper treeline and more highly elevated non-forested
areas to only 10 m to prevent large areas above the forests from being excluded. After visual proof
and deletion of strong outlying points, a final number of 7,081 points for the UT and 5,220 for the
lower treeline (LT) were used for the interpolation of the treeline surfaces applying the natural
neighbor method (Watson, 1992). Subsequently, the vertical distance of the treelines, the area above
and below the treeline were calculated. A buffer of 1000 m around these areas was chosen to
represent the treeline boundary area, because this distance meets the spatial resolution of the Spot
VGT and climate data (Fig. 3).

The distribution of the different ecozones was adapted from Gunin and Vostokova (2005). At several
places the map does not match the position of the landscape elements represented in the remote
sensing data. Thus, these spatial deviations were corrected to the positions of the latter. The
different vegetation units were generalized to the main ecozones (desert, desert steppe, steppe,
forest-steppe, subtaiga, taiga, alpine vegetation). Forests of floodplain areas, which are
hydrologically favored by groundwater, were excluded from this analysis. When forest areas were
found in steppe regions, those parts were changed into forest-steppe. In the upper mountains where
the strong disparity between north-facing slopes with forest and south-facing slopes with steppe
dissipates, the areas with slopes covered by forests in every direction were reclassified as mountain
subtaiga. Subsequently, the mapped forest areas were combined with the ecozones to achieve a
spatial differentiation between forested area and open grassland within the total ecozone (TE) of the
forest-steppe, subtaiga and taiga. These three ecozones comprise the area under investigation in the
present study. In addition, the mapped forest area was combined with digital tree spices maps
provided by the NAMHEM, Ministry of Nature, Environment and Tourism, Mongolia (2009) to receive
spatial tree species data.

Here, the statistical approach to use only one mean value in a period of 15 years (1999-2013) for
every parameter was chosen in order to eliminate annual changes and inter-annual variations, which
derive from phenology and climate variability. Thus, normalized variables representing the mean site
conditions were computed and spatially analyzed. NDVI, temperature and solar radiation are directly
combined to the MGS. Precipitation during the winter season is retained in the soil and additionally
available during the MGS. The vegetation index from SPOT VGT satellite data was used for the time



span from January 1st, 1999 to December 31st, 2013, which originally consists of SPOT-Vegetation 10-
daily NDVI composites (spatial resolution 1 km). These data were aggregated to monthly values using
the maximum value of the three 10 day composites. Monthly NDVI data were further aggregated to
the mean of the growing season from May to September (MGS-NDVI) for the period 1999 to 2013.
We used re-analyzed climate data from the CHELSA dataset with 30 arc sec resolution (approx. 1 km)
(Karger et al., 2016). Monthly data from 1999 to 2013 were averaged to cover the same period as the
MGS-NDVI dataset. While mean growing season temperatures (MGST) were calculated from the
monthly means from May to September, the mean annual precipitation (MAP) represents the
average of the total annual sum of the period from 1999 to 2013. The sum of solar radiation input
(MGSR; Wh m$^{-2}$) for the MGS (day 121-273) was calculated based on STRM-DEM data for 2007 and
was assumed to be relatively constant for the observation period 1999 to 2013.

Up to 3000 random points for both, forest and grassland area in the three ecozones and at the upper
and lower forest boundary were chosen for statistical analysis (Tables 1, 2; Fig. 4). The total number
of random points was reduced for treeline subunits which have only a small spatial distribution to
prevent a too large point density. While the subtaiga is bordering to the meadow-steppe, the lower
treeline seldom occurs in the taiga zone, because the precipitation input in these regions is mostly
high enough for tree growth. For the region of Mongolia, this is true for the large basins and valleys.
Nevertheless, at smaller intermountain basins and smaller valleys, which are rain shadowed by the
surrounding mountains, a lower forest boundary is detectable. When including the isolated lower
treeline values into the interpolation process, the lower treeline surfaces passes the larger valleys
where extensive forest occurs beneath it. These areas are excluded from the treeline analysis.

For each of the three forest-bearing ecozones (forest-steppe, subtaiga, taiga), first, the total area
(total ecozone, TE) is considered, then, the TE is divided into forest (f) and grassland (s) and further
reduced to the 1 km boundary area of both treelines (LT, UT). This categorization leads to 18
ecological subunits, which are analyzed separately. Multiple comparison between means were
calculated with Duncan's multiple range test after testing for normal distribution using SAS 9.4
software (SAS Institute Inc., Cary, North Carolina, U.S.A.). In addition to the mean values, the
standard deviation specifies the variation range of the climate parameters for every subunit. Pearson
and multiple correlation coefficients between NDVI, MAP, MGST, and MGSR were computed as
statistical base for the interpretation of regression trends. Due to the high amount of random points,
it was opposed to perform a t-test because the significance level (p-value) is always <0.05. The
correlations at the level of the TE are used to analyze the controlling climatic conditions and the
environmental range with respect to the ecology of the entire ecozone. In contrast, the treelines
represent boundaries of forest distribution at the ecological limits and it is hypothesized that changes
in climate or environmental conditions at these boundaries lead to an alteration of the treelines.

## 4. Results

### 4.1 Treeline distribution

The actual total area of Mongolian southern boreal forest was estimated at 73,818 km$^2$ (Dulamsuren
et al., 2016). The spatial ratio of forested areas related to the total ecozone areas and in the 1 km
boundaries at the treelines are given in Table 3. While the approximate forest portion is 40 %, low
forest densities occur at all LTs and in the TE of the forest-steppe. Figure 5 shows the forest
distribution, the treelines, the vertical distance of the forest belt, and the area beyond the treelines
in northern Mongolia. The forest area surrounding the Mongolian border was additionally mapped to
receive continuous treeline values crossing the administrative border, but the Siberian region further
to the north was omitted. No treeline continuance is indicated in the southern part of Mongolia due
to missing boreal forests in the desert. The treeline distribution in western Mongolia generally



corresponds to the results from Klinge et al. (2003), who investigated forest distribution in the Altai
Mountains based on topographic maps.
Large areas above the UT occur in the MA, in the southern part of KaM and east of Lake Khuvsgul. In
the KeM areas above the treeline in >2500 m a.s.l. are small. The UTs show a general increase from
2200 m a.s.l. at the Mountains in the North of Uvs Nur and from 1800 m a.s.l. south of Lake Baikal to
2700 m a.s.l. in the southern parts of the MA and the KaM (Fig. 5a). In the southwestern side of the
MA the UT increases steeply from 2100 to 2600 m a.s.l. in a northeastern direction. In the large
mountain systems of the MA and KaM the UT stays in a relative constant altitude between 2400 and
2600 m a.s.l. Northeast of KaM, the UT has an explicit longitudinal direction and a UT depression of
up to 800 m occurs in the basin of the Selenga River. It was verified using the forest cover change
data of Hansen et al. (2013) that the extraordinary low UT in 1800 m a.s.l. is not related to burnt
forest. Large areas below the LT exist in the great basins and along the main river valleys, but they
are also present in the intermountain basins (Fig. 5b). In northern Mongolia, the LT disappears in the
large valleys and forests extend continuously into the valley bottom. However, a distinct LT is still
present in intermountain basins. Concordant with the increasing aridity the LT is generally rising
southward from 1000 to 2500 m a.s.l. in eastern Mongolia. The steep gradient of >1200 m height at
the north- and southwestern edges of the Altai Mountains is due to the enhanced capture of rainfall
at the western ranges of the Altai and the increasing aridity in the MA.
The potential forested area in central Mongolia, which is left between the resulting large areas
beyond the treelines, is small from top-down view. However, the spatial expansion of forests has a
particular vertical component (Fig. 5c). The altitudinal extension of the forest belt reaches its highest
amount of up to 1000 m in the northwestern subtaiga and taiga regions. In the mountain forest-
steppe of the central MA, the western KaM, and in the mountains at Lake Khuvsgul, the altitudinal
extension of forests decreases below 400 m. In the southeastern part of the MA, the UT and LT
converge and the forest belt thins out so that the steppe directly passes over to the alpine zone. Due
to the extraordinary low UT, thin forest belts also occur in the area northeast of the KaM and in the
southwestern part of KeM. This can be related to human impact by wood cutting in a more
populated region. Main precipitation is transported by the westerlies and while the western side of
the Altai Mountains is humid, the dry central MA and the Valley of the Great Lakes, which is located
east of the MA, are directly situated in its rain shadow. This causes an extraordinary high LT and the
small vertical extension of the forest belt in this region (Klinge et al., 2003). The southern side of the
KaM is still arid, but its northern part and particularly the KeM receive more precipitation coming
from the northeast along the Selenga river depression. The tree species composition of the different
ecozones and subunits is given in Figure 6. Siberian larch (*Larix sibirica*) is the dominant tree species
in Mongolia. However, the cedar (*Pinus sibirica*) fraction increases particularly at the UT of the
subtaiga and taiga where the precipitation limit is less important. Additionally, birch (*Betula
platyphylla*), aspen (*Populus tremula*), and pine (*Pinus sylvestris*) trees are occurring at all LTs.
**4.2 Climate parameters of different ecozones**
The zonal statistics for the climate parameters and MGS-NDVI in different ecozones and subunits are
given in Table 1 and the correlation matrix between MGS-NDVI, MAP, MGST, and MGSR is presented
in Table 2. Fig. 4 illustrates the frequency distributions and linear regressions between these
parameters. The average MAP of the TE forests generally increases from 266 mm y$^{-1}$ in the forest-
steppe to 339 mm y$^{-1}$ in the subtaiga and 357 mm y$^{-1}$ in the taiga (Table 1). Due to the hydrological
limitation, the MAP at the LT is lower than the respective average of the TE. This is also true for all
forest subunits at the UTs, where the MAP is about 30 mm y$^{-1}$ lower than the mean average of the TE
forests. This aspect is due to the lower temperatures in higher mountains, which reduce the



evapotranspiration pressure. Interestingly, the average MAP at the UT of the forest-steppe is even
lower than at the LT. However, sites with extremely low MAP below 190 mm y$^{-1}$ (Fig. 4a) must be
related to additional water supply. The grassland has predominantly lower mean values of MAP than
the forests of the corresponding subunit. This general trend inverts at the UTs of the subtaiga and
taiga, while there are nearly equal values at the LTs of the forest-steppe and taiga.
The average MGST in all three TEs are very similar between 11.0 and 11.7 °C. However, the maximum
of 16 °C in the taiga is lower than in the forest-steppe and subtaiga where it is up to 18 °C (Fig. 4b).
While all mean values of MGST at the LTs equate to the TE values, the UTs show frequency maxima
of the MGST between 7.5 and 8.9 °C (Table 1). With the exception of the UT in the subtaiga and taiga,
in all subunits, the grasslands have similar or slightly higher temperatures as the forests of the same
unit. This phenomenon of an inversion of the general trend at the UT of the subtaiga and taiga occurs
simultaneously to the MAP. Here, the grassland is not represented by mountain meadow steppe but
by alpine shrub and meadow vegetation, which is provoked by a cold but more humid climate. The
MGST of all TEs and LTs shows similar frequency distributions with wide value ranges and slightly
higher values at the LTs (Fig.4b). However, the narrow and uniform frequency distributions of all UTs
indicate that the MGST is the main controlling parameter for forests distribution at the UT with an
absolute minimum value of 6 °C. A considerable portion of MGST at the UTs occurs between 10 and
13 °C, which is marginal in the forest-steppe and subtaiga but becomes more important in the taiga.

**4.3 Relationship between climate and NDVI in different ecozones**

The mean values of MGS-NDVI in Table 1 show only slight variation between the ecozones and
subunits. The values increase from forest-steppe to taiga and are higher in the forested area
compared to the grassland of the same subunit. The inverse trends of relation between forest and
grassland of the same subunit, which occur for MAP and MGST at the UT of subtaiga and taiga, do
not exist for the NDVI. The frequency distributions of MSG-NDVI for the subunits in the forest-steppe
are nearly similar but clearly separated in the other ecozones (Fig. 4c). The UTs have the lowest and
the TEs have the highest NDVI values, which is generally due to less favorable ecological site
conditions at the forest boundaries. In Table 2 most of the TEs show good correlations between NDVI
and the climate parameters (r = 0.44-0.71), with an obvious exception of the MAP in the taiga TE.
Linear regressions of the relief parameter MGSR are omitted in Fig. 4, because MGSR is only weakly
correlated to the NDVI in all subunits.
In accordance with the correlation coefficients given in Table 2, the linear fit of the regressions
between MGS-NDVI, MAP, and MGST, which are shown in Fig. 4, illustrates the relationship and
potential susceptibility of the ecozones and corresponding treelines to changes in climatic conditions.
There are mostly low correlations between MGS-NDVI and MAP at most subunits. The only
exceptions are the TE and the LT of the forest-steppe and particular the LT in the forest subunit of
the taiga. However, the gradients of linear regression indicate potential relations between NDVI and
MAP for all LTs and particularly for all subunits in the forest-steppe (Fig. 4a). Both, the correlation
values and the linear regressions between MGS-NDVI and MGST (Fig. 4b) indicate strong
dependencies for all subunits; the UT of the forest-steppe is an exception from this rule, since only
weak correlation was found. However, the steep gradient of the linear regressions at all UTs
accentuates the temperature as the main limiting parameter with increasing influence towards the
taiga. Presupposing that at least precipitation, temperature and solar radiation input control the
vitality of the vegetation and the treeline distribution but with different intensities for every subunit,
the multi-regression correlations between NDVI and MAP, MGST, and MGSR are generally higher.
However, the combination of the two climate parameters MAP and MGST shows the best



correlations with the NDVI, while the combination of all three parameters only leads to a marginal
improvement (Table 2).
The high positive correlations between MAP and MGST and the high negative correlation between
MGST and MGSR in the TE and at the LT of the forest-steppe indicate a specific environmental
interrelation and potential auto-correlation effects between these two climate parameters in the
semiarid climate zone. This is due to the fact that in the forest-steppe the increasing atmospheric
vapor pressure deficit, which results from higher temperatures, must be compensated by more
precipitation, on the one hand, and by less solar radiation input, on the other hand. However, the
weak correlation between MAP and MGST in all subunits of the subtaiga and taiga indicate a climate
independent factor. This is notably attributable to permafrost distribution as a supplemental
ecological parameter, which is not included in our regression models but modifies the soil
hydrological regime. Regression gradients between MAP and MGST of the TEs change from the
strong positive trend in the forest-steppe into a less precipitation-dependent trend in the subtaiga
and then into a negative trend in the taiga (Fig. 4d). The increasing MAP produces more humid
climate in the taiga and makes vegetation vitality in the TE less dependent on precipitation limits.
Low temperatures as zonal climatic parameter become a dominating limit for tree development
towards higher latitudes. Concordant to the transformation of ecological conditions, the
physiological constitution of trees and the tree species composition changes from drought-adapted
to low-temperature adapted but more drought sensitive individuals.
**5. Discussion**
The lower boundaries of the distribution curves (Fig. 4a) and the standard deviation of the MAP
(Table 1) indicate that an approximate MAP of 190 mm $y^{-1}$ can be regarded as the minimum amount
of direct rainfall for tree development in Mongolia. Sites with lower MAP values, occurring in parts of
the forest-steppe, are favored by additional soil water supply from upslope area or melting
permafrost ice, which can support tree growth under these dry conditions  (Dulamsuren et al., 2014).
The annual amount of precipitation is highly varying in the steppes region and the permafrost layer
aids to bridge dry years by accumulating soil water during more humid years (Sugimoto et al., 2002).
The vegetation vitality as expressed by the NDVI is generally lower in the forest-steppe than in the
subtaiga and taiga. This fact proves the extreme ecological limitations of the forest-steppe ecotone.
The recently emerging margins of dead trees around the forest islands are apparently induced by the
trend of increasing temperature, insufficient precipitation, and missing soil water storage from
disappearing discontinuous permafrost.
The proportion between predominant open grassland area and forest islands in the southern forest-
steppe changes towards northern latitudes with the expansion of forest area. In the large valleys of
the taiga and subtaiga in northern Mongolia, where trees are not limited by water scarcity, a LT does
not exist. However, inside the dense woodland of the southern Siberian taiga, the grassland occurs in
intra-mountainous basins where precipitation is extraordinarily low (Dulamsuren et al., 2005; Gunin
et al., 1999; Hilbig, 1995) and thus a LT is present. The high correlation of the detected LTs to MAP in
the taiga ecozone proves the more natural than human-induced existence of this forest distribution
boundary and its susceptibility to aridification. This conclusion is supported by ecophysiological,
dendrochronological, and palynological studies from such areas (Dulamsuren et al., 2009a; 2010b;
Schlütz et al., 2008).
The correlation between NDVI and MGST at the UT is strong in the taiga and subtaiga regions (Table
2). At the UT of the forest-steppe region, precipitation is a concurrent limiting factor at higher
elevations. While a MGST of 6 °C tends to be the general minimum temperature for tree growth in
the study area, at some places at the UT of the subtaiga, trees occur at MGST as low as 4 °C (Fig. 4b).



At these locations, the low MGST is associated with high MAP of roughly 350 mm y$^{-1}$ (Fig. 4d). At the
low temperature range between 6-8 °C, the linear regressions between MAP and MGST at the UT
show that, at these cold sites, different MAP conditions exist simultaneously for the different
ecozones (Fig. 4d). In the forest-steppe at 6 °C MGST, MAP is approximately 200 mm y$^{-1}$, whereas it
amounts to c. 320 mm y$^{-1}$ in the subtaiga and 400 mm y$^{-1}$ in the taiga. This combination between
both low precipitation and temperature is most extreme at the LT of the forest-steppe. In the range
of 6-8 °C MGST, MAP tends to be below the tree growth minimum of 190 mm y$^{-1}$, which emphasizes
again the impact of permafrost, as the permafrost is also associated with low temperatures.
Differing frequency distributions show that the NDVI at the UT and LT is generally lower than in the
TEs of the taiga and subtaiga, except for the forest-steppe (fig 4c). The low NDVI values indicate low
vegetation vitality. This suggests that forests composing treelines in the taiga and the subtaiga and
the complete forest-steppe ecotone are exposed to physiological stress. Reports of increased
drought stress, reduced stemwood formation, reduced forest regeneration and increased tree
mortality especially in the *Larix sibirica*-dominated forest-steppe ecotones of Inner Asia support this
conclusion (Dulamsuren et al., 2010a; 2010b; 2013; Liu et al., 2013). Future climate warming with
increased summer drought will change dominating tree species from *Larix sibirica* to *Pinus silvestris*
in places of the forest-steppe (Dulamsuren et al., 2009b). Forests in the taiga receive generally more
precipitation and thus have thus developed higher stand densities and are also home to more water-
demanding dark taiga tree species (Dulamsuren, 2004; Dulamsuren et al. 2010a).

**6. Conclusion**

Using high resolution remote sensing and climatic data enables to specify the climatic framework of
the three forest-bearing ecozones in Mongolia and to indicate additional factors for vegetation
growth. Differing tendencies in the NDVI distribution between forest and grassland of the same
subunits, which are mainly controlled by different photosynthetic activity, vegetation density, and
seasonal growth, were also found. However, with respect to the small-scale variation of the
vegetation and the ground resolution of the NDVI-data a spatial overlap producing mixed data values
cannot be totally avoided.
In summary, the ecological relationship between climatic parameters and forest or treeline
distribution can be verified by the NDVI as indicator for vegetation vitality. But site conditions like
permafrost distribution, soil parameters and hydrology play an important role for vegetation vitality,
too. The statistical results presented in this work are adequate to be used for projection and
modeling of potential forest development on the one hand or for vegetation based refinement of
climate data on the other.
We conclude that rising temperatures induced by global warming will finally lead to less tree vitality
and forest degradation in the forest-steppe, subtaiga, and taiga as well. Even a simultaneous increase
of precipitation will be consumed by more evapotranspiration. The observed recent increase of
forest greening indices from remote sensing data and stemwood increment found in several places is
combined to increasing summer temperature but also promoted by additional soil water supply from
melting permafrost. However, disappearing permafrost and increasing drought stress on less
drought-tolerant trees can subject hazardous distortion to forests in the future. For all LTs and for
the TE of the forest-steppe, rising temperatures will lead to tree mortality, the reduction of forested
area, and shifting of the LTs. Even a contemporaneous increasing precipitation cannot totally
compensate for the disappearing permafrost because this leads to insufficient soil water during dry
years. The existence of the widespread dead tree margins at forest islands proves that this trend is
already ongoing concurrent to the temperature increase during the last decades. Unadapted trees



suffering from drought stress are increasingly vulnerable to insect calamities and mortality and less
resistance to many of the recent forest fires.
Research on NDVI trends and climate change in Mongolia is often lacking detailed spatial separation
of the different ecozones. Every ecozone has its own temporal and ecological environment, which
produces different trends in remote sensing derived vegetation indices. The local climatic and soil
site conditions induce the growth of physiologically adapted trees and tree species. A climatic change
will lead to more or less vitality but in the limited physiological range of the individuals. Forest
dynamic and forest development from the biological point of view means change in the vegetation
structure and biodiversity, which cannot be exclusively modeled by greening indices (Miao et al.,
2015; Poulter et al., 2013). For future investigation on vegetation development in relation to climate
trends, it is strictly necessary to consider the ecological transitions. It was shown, that the creation of
a detailed landscape stratification and of small scaled ecological classifications can assist to
incorporate spatial and temporal transitions of vegetation units in environmental modelling or
projection.
**Acknowledgements**
The authors would like to thank the US Geological Survey for making the satellite data freely
available for scientific research. We acknowledge support by the Open Access Publication Funds of
Göttingen University. We very specially thank our colleague Dr. Jan Degener for his scientific support
in data processing and valuable discussion.
This open-access publication was funded by the University of Göttingen.



Tables:
Table 1: Arithmetic mean ± standard deviation of different climate parameters (MAP: Mean Annual
Precipitation, MGST: Mean Growing Season Temperature, MGS-NDVI: Mean Growing Season
Normalized Differentiated Vegetation Index) and vegetation units (Subunits are TE: Total Ecozone,
LT: Lower Treeline, UT: Upper Treeline, s: portion of grassland, f: portion of forest). Within one row,
mean values sharing a common uppercase letter, do not differ significantly (P≤0.05, Duncan's
multiple range test, $df_{model}$ = 2).Within one subunit (forest-steppe, subtaiga, taiga), mean values
sharing a common lowercase letter, do not differ significantly (P≤0.05, Duncan's multiple range test,
$df_{model}$ = 5, 13295).

| Subunit | Forest-steppe | Subtaiga | Taiga |
|---|---|---|---|
| MAP (mm y$^{-1}$) | | | |
| TE$_f$ | 266±62 Aa | 339±70 Ba | 357±69 Ca |
| TE$_s$ | 256±63 Ab | 309±68 Bbe | 331±73 Cb |
| LT$_f$ | 251±60 Ac | 294±60 Bc | 292±56 Bc |
| LT$_s$ | 253±62 Abc | 286±57 Bd | 290±53 Bc |
| UT$_f$ | 231±52 Ad | 305±72 Be | 333±80 Cbd |
| UT$_s$ | 227±54 Ae | 314±73 Bb | 339±80 Cd |
| MGST (°C) | | | |
| TE$_f$ | 11.0±2.1 Aa | 11.7±2.3 Ba | 11.1±1.4 Ca |
| TE$_s$ | 11.6±2.5 Ab | 11.7±2.7 Ba | 11.1±1.7 Ca |
| LT$_f$ | 11.5±2.2 Ab | 12.1±2.6 Bb | 11.5±1.7 Ab |
| LT$_s$ | 12.1±2.3 Ac | 12.8±2.4 Bc | 11.7±1.6 Cc |
| UT$_f$ | 8.4±0.8 Ad | 7.9±1.2 Bd | 8.9±1.3 Cd |
| UT$_s$ | 8.4±0.9 Ad | 7.5±1.2 Be | 8.5±1.3 Ce |
| MGS-NDVI | | | |
| TE$_f$ | 0.51±0.08 Aa | 0.60±0.08 Ba | 0.63±0.06 Ca |
| TE$_s$ | 0.47±0.08 Ab | 0.55±0.09 Bb | 0.55±0.09 Cb |
| LT$_f$ | 0.46±0.08 Ab | 0.54±0.08 Bc | 0.58±0.09 Cb |
| LT$_s$ | 0.44±0.08 Ac | 0.51±0.08 Bd | 0.55±0.08 Cc |
| UT$_f$ | 0.44±0.06 Ac | 0.47±0.07 Be | 0.51±0.09 Cd |
| UT$_s$ | 0.42±0.07 Ad | 0.44±0.08 Bf | 0.47±0.09 Ce |






Table 2: Correlation matrix showing Pearson and multiple correlation coefficients (r) between NDVI,
climate, and relief parameters for different ecozones and subunits. (MAP: Mean Annual
Precipitation, MGST: Mean Growing Season Temperature, MGS-NDVI: Mean Growing Season
Normalized Differentiated Vegetation Index, MGSR: Mean Growing Season Solar Radiation Input,
subunits are TE: Total Ecozone, LT: Lower Treeline, UT: Upper Treeline, s: portion of grassland, f:
portion of forest)

| Subunit | Forest-steppe | Subtaiga | Taiga | Forest-steppe | Subtaiga | Taiga | Forest-steppe | Subtaiga | Taiga | Forest-steppe | Subtaiga | Taiga |
|---|---|---|---|---|---|---|---|---|---|---|---|---|
| | MGS-NDVI / MAP | | | MGS-NDVI / MGST | | | MGS-NDVI / MGSR | | | | | |
| TE f | 0.58 | 0.44 | 0.22 | 0.49 | **0.62** | 0.55 | -0.15 | -0.24 | -0.09 | | | |
| TE s | 0.57 | 0.38 | 0.19 | 0.49 | 0.55 | 0.57 | -0.26 | -0.17 | -0.18 | | | |
| LT f | 0.53 | 0.33 | 0.51 | 0.56 | 0.52 | **0.60** | -0.09 | -0.20 | -0.18 | | | |
| LT s | 0.55 | 0.39 | 0.39 | **0.61** | 0.52 | 0.46 | -0.29 | -0.29 | -0.30 | | | |
| UT f | 0.34 | 0.11 | 0.34 | 0.31 | 0.59 | **0.71** | 0.22 | 0.19 | 0.08 | | | |
| UT s | 0.42 | 0.10 | 0.33 | 0.25 | 0.55 | **0.66** | 0.15 | 0.17 | 0.08 | | | |
| | MGS-NDVI / MAP ; MGSR | | | MGS-NDVI/MGST ; MGSR | | | MGS-NDVI / MAP ; MGST | | | MGS-NDVI / MAP ; MGST ; MGSR | | |
| TE f | 0.58 | 0.47 | 0.24 | 0.51 | **0.62** | 0.56 | **0.62** | **0.71** | **0.64** | **0.63** | **0.72** | **0.65** |
| TE s | 0.58 | 0.41 | 0.26 | 0.50 | 0.56 | 0.58 | **0.62** | **0.67** | **0.68** | **0.63** | 0.67 | **0.68** |
| LT f | 0.53 | 0.36 | 0.52 | 0.59 | 0.52 | 0.60 | **0.60** | 0.56 | **0.72** | **0.63** | 0.57 | **0.72** |
| LT s | 0.56 | 0.43 | 0.42 | 0.61 | 0.52 | 0.50 | **0.64** | 0.58 | 0.57 | **0.65** | 0.58 | 0.58 |
| UT f | 0.36 | 0.24 | 0.35 | 0.37 | **0.60** | 0.71 | 0.43 | **0.62** | **0.74** | 0.45 | **0.64** | **0.75** |
| UT s | 0.42 | 0.21 | 0.34 | 0.30 | 0.56 | **0.66** | 0.47 | 0.58 | **0.69** | 0.47 | 0.59 | **0.69** |
| | MAP / MGSR | | | MGST / MGSR | | | MAP / MGST | | | | | |
| TE f | -0.23 | -0.16 | 0.05 | -0.54 | -0.48 | -0.29 | 0.50 | 0.16 | -0.18 | | | |
| TE s | -0.29 | -0.05 | 0.01 | **-0.67** | -0.42 | -0.26 | 0.47 | 0.00 | -0.28 | | | |
| LT f | -0.22 | -0.21 | -0.16 | -0.46 | -0.47 | -0.24 | **0.63** | 0.23 | 0.20 | | | |
| LT s | -0.37 | -0.29 | -0.41 | -0.58 | -0.44 | -0.23 | **0.65** | 0.24 | 0.12 | | | |
| UT f | 0.25 | -0.18 | 0.01 | 0.05 | 0.11 | 0.04 | 0.12 | -0.11 | 0.18 | | | |
| UT s | 0.20 | -0.10 | -0.05 | 0.00 | 0.12 | 0.16 | 0.11 | -0.15 | 0.17 | | | |



Table 3: Spatial ratios of forest area (f) and total area of different ecozones and corresponding
treelines. (TE: Total Ecozone, LT: Lower Treeline, UT: Upper Treeline)

| area km² | TE | TE_f | %_f | LT | LT_f | %_f | UT | UT_f | %_f |
|---|---|---|---|---|---|---|---|---|---|
| Forest-steppe | 62,678 | 17,983 | 28.7 | 17,275 | 3,894 | 22.5 | 3,525 | 1,822 | 51.7 |
| Subtaiga | 87,648 | 38,747 | 44.2 | 7,558 | 2,135 | 28.2 | 3,168 | 1,341 | 42.3 |
| Taiga | 31,710 | 17,088 | 53.9 | 1,234 | 401 | 32.5 | 949 | 495 | 52.2 |
| Sum | 182,036 | 73,818 | 40.6 | 26,067 | 6,430 | 24.7 | 7,642 | 3,658 | 47.9 |






Figures:

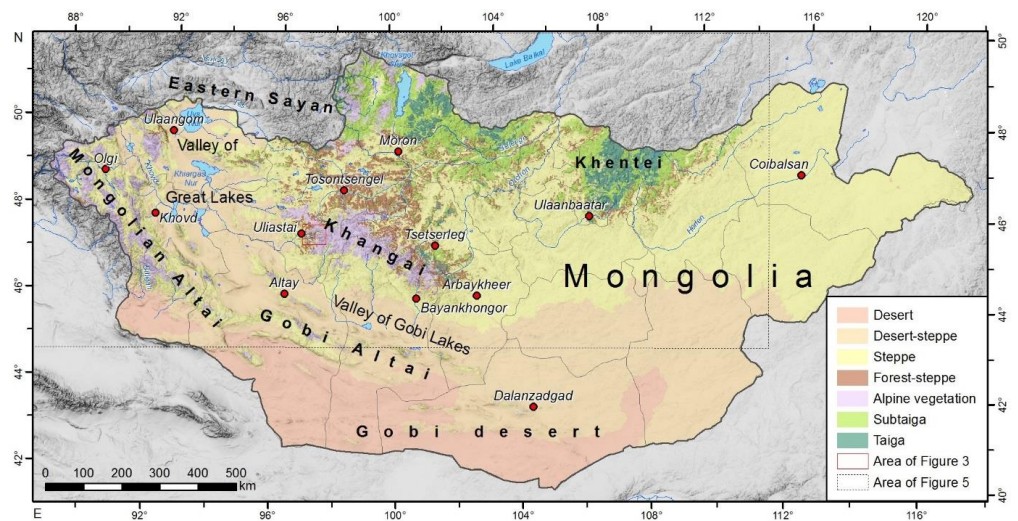

Figure 1: The vegetation zones of Mongolia (modified from Gunin and Vostokova, (2005) and Landsat
8 supervised classification).





Figure 2: Processing workflow for treeline delineation, NDVI and climate analysis.





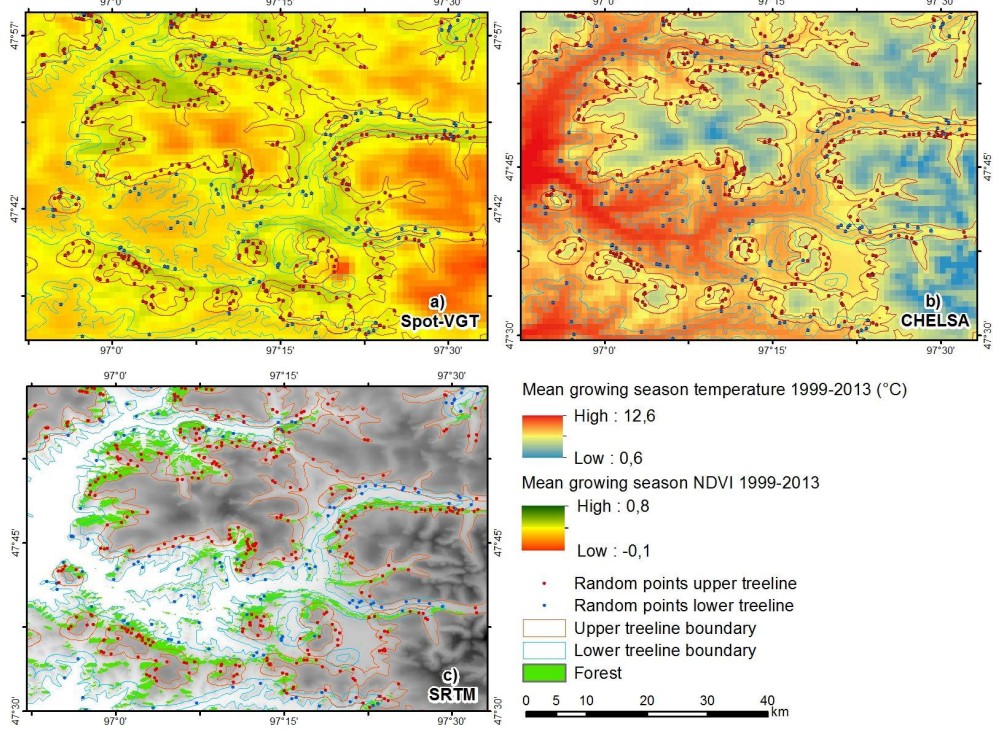


Figure 3: Examples for the spatial resolution of the different data: (a) mean growing season NDVI
1999-2013, (b) mean growing season temperature 1999-2013, (c) upper and lower treeline boundary
from Landsat and SRTM data.




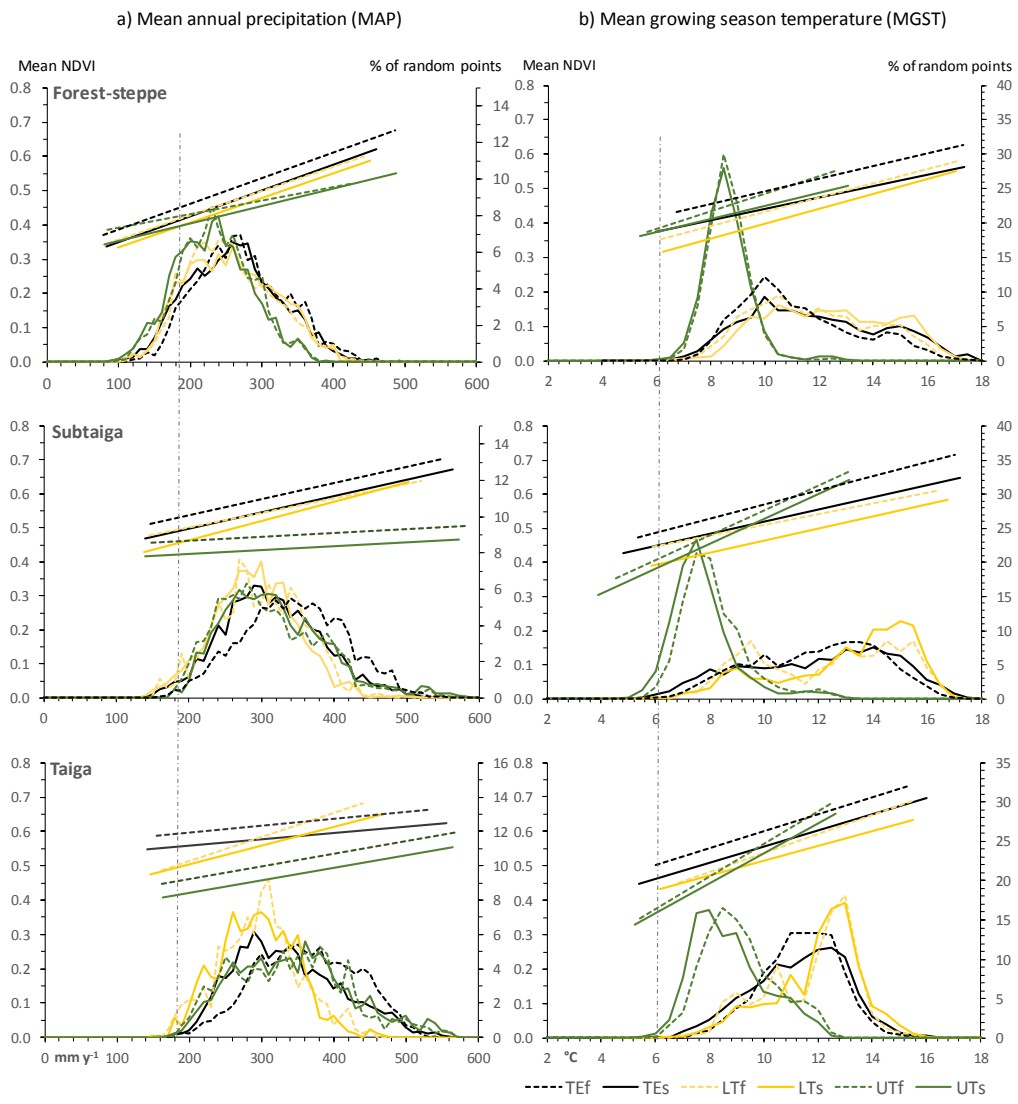

Figure 4 (continued)



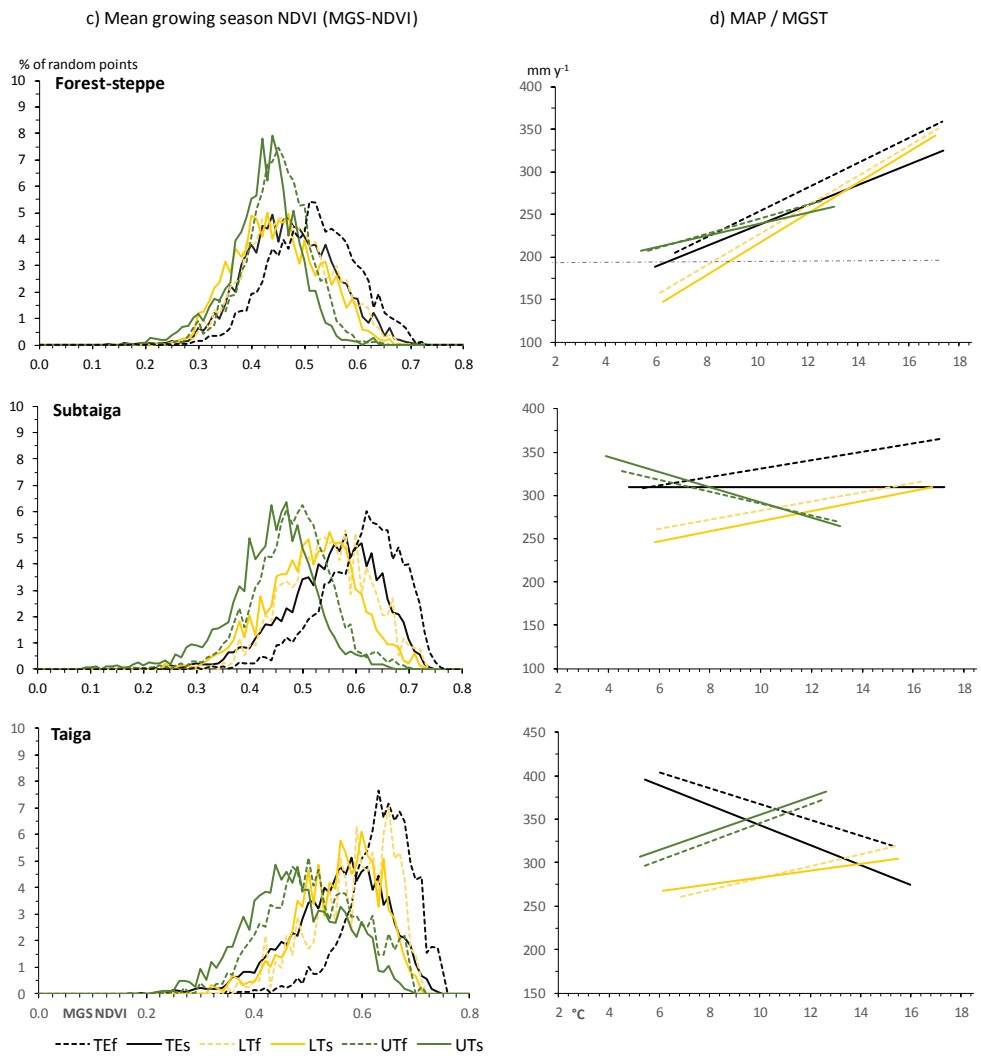


Figure 4: Mean annual precipitation (MAP) and mean air temperature during the growing season
(MGST) related to the mean growing season NDVI of random points of different ecozones (values
averaged for the investigation period 1999-2013). The straight lines are representing the linear
regressions between climate parameters and NDVI. The distribution curves represent the frequency
of random points (%). Dashed lines represent forest values (f); continuous lines represent grassland
values (s); yellow colors represent lower treeline values (LT); green color represents upper treeline
values (UT) and black colors represent the total ecozone values (TE). Vertical grey dashed lines
indicate the deduced minimum values for tree growth.






Figure 5: Treeline distribution maps of Mongolia: (a) upper treeline, (b) lower treeline, (c) vertical
distance between upper and lower treeline (a.a.t. = area above the upper treeline, a.b.t. = area
beneath the lower treeline, f.b.l.t. = forest below the lower treeline)







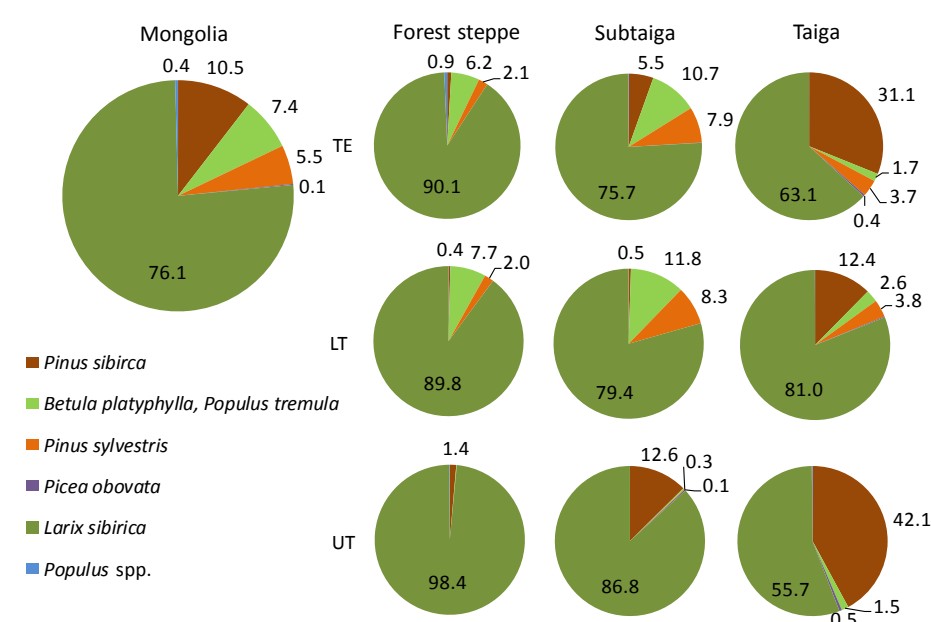


Figure 6: Tree species composition in the boreal zone of Mongolia and in the subunits of three
different ecozones. The tree species distribution was adapted from Data provided by NAMHEM,
Ministry of Nature, Environment and Tourism, Mongolia (2009)



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
