# Peer review of "Climate effects on vegetation vitality at the treeline of boreal forests of Mongolia"

_Biogeosciences, 2017_

## Referee Comment (RC1) · U. Schickhoff (Referee) · 8 Dec 2017

The paper is of considerable interest to a broad readership of Biogeosciences since it is one of very few studies that focus on the relationship of forest distribution and treeline positions with climatic parameters in Mongolia. Especially the combination with remote sensing indices (NDVI) had been rather neglected so far. The paper is an important original contribution, and I recommend it for publication after - from my perspective - necessary revisions. In my view, the usage of the term 'ecozone' in the paper (in the title, throughout the text, in the figures) is not appropriate. The term 'ecozone' is associated with large-scale units (biomes) such as humid mid-latitudes, dry mid-latitudes etc. I believe it is confusing to use it for small-scale units as in this paper. The authors should apply a consistent, generally accepted terminology for habitats / vegetation for-

mations along horizontal and altitudinal zonations (see below). Without explicitly saying so the authors suggest that treeline positions (upper and lower treelines) are in accordance with present climatic conditions. This must not necessarily be the case. Most treelines are in a process of climate tracking, and lag behind climatic changes, in particular when those changes take place very fast. Human impact on treelines is only briefly touched in the paper. I would like to have stressed by the authors the influence of e.g. climate history, vegetation history and historical human impact, and to what extent these factors might influence the results of this study. The manuscript needs language editing in many lines, I suggest to involve a native speaker. Specific remarks as follows: - line 54: ...strongly varies in space and time... - line 61-62: soil temperature, soil moisture and soil nutrients might also play a role - line 64-65: usage of the term 'ecozone' is confusing. Regarding altitudinal zonation I suggest to use the term 'zone' or 'belt' (alpine zone or alpine belt), regarding horizontal zonation I suggest to use the term 'habitat' or 'zone' or another term since the term 'ecozone' is associated with large-scale units (biomes) such as humid mid-latitudes, dry mid-latitudes etc. - line 67: altitudinal zones are not biomes, but zones or belts - line 85: no comma after et al. - line 103: either no comma before which or comma after Spot VGT) - line 105: see line 103 - line 128-130: language editing - line 144: trends of instead of trends for - line 166: showed the NDVI to be well usable.... - line 167: tree biomass of Mongolian forests - line 176: climatically restricted? I suggest to rewrite: ...'is delimited by a constellation of climatic threshold values' - line 177-178: reflect climate-ecological relationships and limitations - line 195: highly continental semi-humid - line 196: with little snowfall - line 205: ...are arranged in characteristic sequences along latitudinal and altitudinal gradients - line 206: obovata kursiv - line 207: selectively? Please rewrite this sentence - line 211: intra-montane basins - line 217: the terms playas and takirs should be explained - line 222: forest management - line 226/227: Please explain the increasing fire susceptibility - line 241: Fig. 2 - line 244: language editing - line 259: In the upper elevational zones? - line 265: tree species maps - line 268-270: Using this approach the authors should be aware of and should point out that this is a simplification since the plant species respond to inter-annual variations and extreme values; plant species do not respond to mean values - line 293-294: language editing - line 298: multiple comparisons - line 306-308: Alteration of treelines requires successful recruitment of tree species. The authors should be aware of the fact that bioclimatic requirements of seedlings and saplings might deviate to a considerable extent from those of adult trees - line 332: intermontane basins - line 335: language editing - line 340-341: language editing - line 344: language editing - line 347-348: language editing - line 380: blank space - line 381: forest distribution or forest stand distribution - line 433ff: Ulmus trees along water courses in the steppes should also be mentioned here - line 447: intramontane basins - line 472-473: language editing - line 474: 2x thus - line 474-475: hygrophilous instead of water-demanding - line 478: which additional factors? - line 479: results instead of tendencies - line 496: language editing - line 507: Climatic change will lead. . .. - line 509: Forest dynamics - line 510: modelled - Fig. 5: Map legend: there is no reference to the black line (not all of the readers are familiar with the borders of Mongolia) - Fig. 6: Legend: Pinus sibirica

---

## Referee Comment (RC2) · Anonymous Referee #2 · 15 Dec 2017

The manuscript by Klinge et al intends to quantify the influence of climate variables on the distribution of different land cover characteristics in Mongolia and particularly focusses on the upper and lower treeline. A land cover classification is conducted based on remote sensing data for the period 1999-2013 and the position of treelines is derived by means of a complex interpolation strategy. Climate statistics (based on the recently established CHELSEA data set) are shown for each land cover unit

Firstly I would like point out, that I am not a Biogeo-Scientist. Thus I review the manuscript from a technical and statistical point of view. In general, the target of the study is timely, since Central and High Aisan forests are highly affected by climate change and remote sensing data analyses of treelines are rare. However, in my opinion, the manuscript requires some major revisions before potential publications. In the

following I will summarize major concerns without going into detail:

1) Manuscript Structure and Terminology: The structure of the text is partly confusing and a clear separation into introduction, data, methods, results and discussion is not clear. I recommend to shorten the manuscript (and avoid repetitions), particularly the introduction, and to focus on information, which is relevant for the remote sensing based analysis. Further, the methods and results include a lot of additional information, which is not supported by the study. Please clearly separate between data, methods and results of your study and other relevant studies (which can be reviewed in the introduction or the discussion). Further some of the figures are only very briefly mentioned (especially 6 and 3). Please check, if these are needed. In general the figures are in a confusing order which does not follow the structure of the manuscript. Finally the language of the manuscript is partly unclear, misleading or imprecise. E.g. the term "trend" means change with time (I feel the authors often mean "spatial gradient"). The same applies for terms like "decreasing" or "increasing", which indicate temporal change (and not spatial variability). The term "relief parameter" should be changed to "terrain parameter" throughout the manuscript.

2) Land Cover Classification: The paragraph on the classification algorithm is very short. Please give more information on the methods (how many training regions are used, how accurate is the classification, please also quantify the uncertainty)

3) Data: Particularly the climate data are only rudimentarily described. Please give more information on the generation and quality of the data set. This is particularly important for precipitation. Personally I am not sure if the data set is able to reproduce the elevational gradient of precipitation, which appears to be an important trigger of the lower treeline. I suggest to show precipitation and temperature maps and (if possible) compare seasonal and annual data with available station data for a rough quality assessment. Further the generation of solar radiation data is not clear to me? Is it solely based on the DEM or is any atmospheric information included?

4) Ecozones and land cover classes: From my understanding, the larger scale "eco-zones" are driven by climatic conditions (just like the land cover classes derived from remote sensing data). A habitat of a particular vegetation type (class) should actually occupy a certain climatic niche (regardless of the large scale "ecozone"). Please clarify why "ecozones" were used to separate the data. Further, the results indicate different statistical relationships between treeline patterns and climate variables in different zones (inverted relationships, e.g. l.369ff). I wonder if there could be any ecological mechanism behind or whether this is rather a statistical artefact (especially since there is only one ecozone, which is mainly covered by forest).

5) Statistical Methods: The statistical methods for the analysis of climate-landcover relationships are parametrical, i.e. they require normally distributed input data (spatial data depend on the terrain and are certainly not normal distributed). This is relevant for the assessment of significant differences (Tab. 1) and also for the linear regressions (Fig. 4). I feel, the use of the regression is appropriate (since only the direction is discussed in the text), however, limitations should be clearly stated. Table 1 could be shown as boxplots and significance testing should be avoided.

6) Spatial Interpolation of treeline elevation: The treeline is (as the analysis shows) highly influenced by local scale climate conditions. Thus the spatial interpolation seems to be misleading to me. I recommend to illustrate the treelines as polygons. Further the potential anthropogenic influence on the treeline location should be elaborated. Further in Table 3 spatial ratios of forest cover are calculated for the lower and upper treeline. I do not understand, what these ratios mean (the treeline is actually not an area, but an ecological border)?

7) Treelines and climate change: The authors use the spatial pattern of treelines and spatial variations of NDVI in order to discuss the potential impact of climate change on tree growth. For me this link is not trivial and should be better investigated. I recommend to also analyze the temporal variability of NDVI, e.g. with respect to warm, cold, dry or wet years during the observational period. Further, the manuscript contains

very little information about climate change scenarios (and potential changes of temperature and precipitation) for Central Asia. This would be an important basis for the discussion. Once again, please focus on results, which are really supported by your study! E.g. sentences in l. 491/492 or 498/499 (but also others) are very speculative and not proven by the analysis or by literature.

---

## Author Comment (AC1) · 11 Jan 2018

In the reworking procedure, we will first correct your detailed proposals in the manuscript and then work on the comments of the 2nd referee. Doing it that way, some comments of the 2nd referee will already be solved, but in addition larger parts of the manuscript may be rearranged afterwards.

*Ref.: In my view, the usage of the term 'ecozone' in the paper (in the title, throughout the text, in the figures) is not appropriate. The term 'ecozone' is associated with large-scale units (biomes) such as humid mid-latitudes, dry mid-latitudes etc. I believe it is confusing to use it for small-scale units as in this paper. The authors should apply a consistent, generally accepted terminology for habitats / vegetation formations along horizontal and altitudinal zonations (see below).*

Rep.: You are right with the definition of the term "ecozones" and we will try to change the terms in the manuscript as you propose. At least we are focusing on the forest ecosystems, so we will change the term "ecozone" to "type of boreal forest" and the class "Total ecozone" to "Total ecosystem unit" throughout the entire manuscript.

*Ref.: Without explicitly saying so the authors suggest that treeline positions (upper and lower treelines) are in accordance with present climatic conditions. This must not necessarily be the case. Most treelines are in a process of climate tracking, and lag behind climatic changes, in particular when those changes take place very fast. Human impact on treelines is only briefly touched in the paper. I would like to have stressed by the authors the influence of e.g. climate history, vegetation history and historical human impact, and to what extent these factors might influence the results of this study.*

Rep.: OK, you are right. We inserted more information about treeline development and its ecology.

*Ref.: The manuscript needs language editing in many lines, I suggest to involve a native speaker.*

Rep.: Anyway, the English language needs always an improvement. The last version will be checked by a (nearly) native speaker. However, we count on the excellent English language editing, which may finally be done by Copernicus before publishing.

Specific remarks as follows:

*- line 54: ...strongly varies in space and time...*          -done

*- line 61-62: soil temperature, soil moisture and soil nutrients might also play a role*

    - We incorporated this fact with an additional sentence

*- line 64-65: usage of the term 'ecozone' is confusing. Regarding altitudinal zonation I suggest to use the term 'zone' or 'belt' (alpine zone or alpine belt), regarding horizontal zonation I suggest to use the term 'habitat' or 'zone' or another term since the term 'ecozone' is associated*

*with large-scale units (biomes) such as humid mid-latitudes, dry mid-latitudes etc.*

- We changed the sentence and will now consequently use the terms "zone" for horizontal and "belt" for altitudinal zonation

*- line 67: altitudinal zones are not biomes, but zones or belts* - changed

*- line 85: no comma after et al.*

– not changed, because this kind of formatting is demanded by BIOGEOSCIENCE and automatically processed by CITAVI

*- line 103: either no comma before which or comma after Spot VGT)* - changed

*- line 105: see line 103* - changed

*- line 128-130: language editing* – The sentence is not necessary and completely deleted

*- line 144: trends of instead of trends for* - done

*- line 166: showed the NDVI to be well usable....* - done

*- line 167: tree biomass of Mongolian forests* - done

*- line 176: climatically restricted? I suggest to rewrite: ...'is delimited by a constellation of climatic threshold values'* - That is good, changed

*- line 177-178: reflect climate-ecological relationships and limitations* - changed

*- line 195: highly continental semi-humid* - changed

*- line 196: with little snowfall* - changed

*- line 205: ...are arranged in characteristic sequences along latitudinal and altitudinal gradients*

- changed

*- line 206: obovata kursiv* - changed

*- line 207: selectively? Please rewrite this sentence* - changed to: .. locally as mountain taiga ..

*- line 211: intra-montane basins* - changed

*- line 217: the terms playas and takirs should be explained*

– I preferred to delete the sentence, because this information is not really necessary

*- line 222: forest management* - changed

*- line 226/227: Please explain the increasing fire susceptibility*

- explained by climate warming, permafrost retreat, and insect calamities

*- line 241: Fig. 2* - not changed, because directly naming the figure

*- line 244: language editing* - changed and hopefully better described

*- line 259: In the upper elevational zones?* - changed

*- line 265: tree species maps* - changed

*- line 268-270: Using this approach the authors should be aware of and should point out that this is a simplification since the plant species respond to inter-annual variations and extreme values; plant species do not respond to mean values*

- We mentioned this problem following your words; however, in a next step of research it would be interesting to investigate if it possible to detect single extreme values in the data, which may play a significant rule for limiting tree distribution. But first it needs to establish the multidata analysis like shown here, before to go to deep into detailed analysis, while the accuracy of the base database does not fit the research problem.

*- line 293-294: language editing* — changed, sentence shortened

*- line 298: multiple comparisons* — changed

*- line 306-308: Alteration of treelines requires successful recruitment of tree species. The authors should be aware of the fact that bioclimatic requirements of seedlings and saplings might deviate to a considerable extent from those of adult trees*

- You are right. That this is true, we could see during our last fieldwork in Mongolia. We wanted to examine why there is such a bad rejuvenation for larch trees like observed 3 years ago. Now after three rainy and humid summers we found extensive succession and even larch seedlings inside the steppe. However, from the remote sensing point, we can only detect adult trees and forests, which must have had sufficient environmental conditions to survive for a longer period. At the end of this chapter method, we inserted a complete new paragraph, where we described the ecological problems of forest distribution, human impact, and the technical limits of the investigation presented here.

*- line 332: intermontane basins* — changed

*- line 335: language editing* — changed

*- line 340-341: language editing* — changed

*- line 344: language editing* — changed

*- line 347-348: language editing* — changed

*- line 380: blank space* — changed

*- line 381: forest distribution or forest stand distribution* — changed

*- line 433ff: Ulmus trees along water courses in the steppes should also be mentioned here*

Ulmus trees play a minor role in our investigation, because they occur at water-favored places near river and in the basins. Therefore their occurrence is less climate depend and also the basin region were excluded from treeline analysis. However, we inserted the Ulmus trees in the introduction to the Study area.

*- line 447: intramontane basins* — changed

*- line 472-473: language editing* — changed

*- line 474: 2x thus* — changed

*- line 474-475: hygrophilous instead of water-demanding*

not changed because trees are not specific hygrophilous species

*- line 478: which additional factors?*

- We can only assume what the additional factor may be: permafrost, water from upper slope,… The sentence was changed to the meaning that we can identify the position but not the specific ecological exception of extraordinary forest stands.

*- line 479: results instead of tendencies* — changed

*- line 496: language editing* — changed

*- line 507: Climatic change will lead: : :.* — changed

*- line 509: Forest dynamics* — changed

*- line 510: modelled*          - changed

*- Fig. 5: Map legend: there is no reference to the black line (not all of the readers are familiar with the borders of Mongolia)*      - changed

*- Fig. 6: Legend: Pinus sibirica*          - changed

---

## Author Comment (AC2) · 11 Jan 2018

Rep.: We did not perform a complete land cover classification. All the ecological zones we used were gained from maps published by Gunin and Vostokova (2005). These maps are based on longtime field research from Mongolian and Russian botanists. Although there are also landcover maps existing, which were produced by remote sensing classification, we did not use them, because these data are more imprecise than the latter and mostly they do not fit to the question of vegetation zones of our investigation. The only classification process we have done here, is to differentiate between forest or no forest in every single Landsat image. However, this was the first step before manually reworking the previously classified forest polygons by visually checking every Landsat image.

*Ref.: (…) However, in my opinion, the manuscript requires some major revisions before potential publications. In the following I will summarize major concerns without going into detail:*
*1) Manuscript Structure and Terminology: The structure of the text is partly confusing and a clear separation into introduction, data, methods, results and discussion is not clear. I recommend to shorten the manuscript (and avoid repetitions), particularly the introduction, and to focus on information, which is relevant for the remote sensing based analysis. Further, the methods and results include a lot of additional information, which is not supported by the study. Please clearly separate between data, methods and results of your study and other relevant studies (which can be reviewed in the introduction or the discussion).*

Rep.: We have rearranged many parts of the text to receive a more consistent structure. Repetitions and less relevant information were mostly deleted. We also tried to shorten the text. However, we had to insert more detailed description where our statements were unclear.
With respect to the journal's scientific orientation the focus of our manuscript is a bio-geoscientific approach to the treeline in Central Asia by a GIS-analysis of different indicators and remote sensing is only one part of it.

*Ref.: Further some of the figures are only very briefly mentioned (especially 6 and 3). Please check, if these are needed.*

Rep.: Figure 3 it is very important to give the reader a visual idea how the different dataset spatially fit to each other and how the treeline points and the forest boundary was mapped. We inserted more relations to Fig. 3 in the text where it was necessary.
Although figure 6 would make sense to describe the tree species compositions at different subunits, we will delete this information totally from the manuscript. Maybe there is a chance to publish it as supplementary material.

*Ref.: In general the figures are in a confusing order which does not follow the structure of the manuscript.*

Rep.: You are right. We put Figure 4 backward, so first the treeline maps appear and afterwards the statistical discussion.

*Ref.: Finally the language of the manuscript is partly unclear, misleading or imprecise. E.g. the term "trend" means change with time (I feel the authors often mean "spatial gradient").*

Rep: You are right; we have changed the term into gradient where we are not talking about temporal developments

*Ref.: The same applies for terms like "decreasing" or "increasing", which indicate temporal change (and not spatial variability).*

Rep: You are right, we have changed the terms into rise, ascending, enlarging and descending, reducing, where we are talking about spatial patterns.

*Ref.: The term "relief parameter" should be changed to "terrain parameter" throughout the manuscript.*

Rep: You are right, we have changed the term

*Ref.: 2) Land Cover Classification: The paragraph on the classification algorithm is very short. Please give more information on the methods (how many training regions are used, how accurate is the classification, please also quantify the uncertainty)*

Rep.: The only classification process we have done here, is to differentiate between forest or no forest individually for every Landsat image. However, this was the first step before manually editing the previously produced forest polygons by visually checking every Landsat image. We compared the mapped forest area with air photos in the basemaps provided by ArcGIS and corrected them where necessary. Producing a map in that way does not enable to create a reliable confusion matrix.
The Land cover classification was adapted from the Ecosystems Atlas of Mongolia (Gunin and Vostokova, 2005), which we now precisely state in the manuscript.

*Ref.: 3) Data: Particularly the climate data are only rudimentarily described. Please give more information on the generation and quality of the data set. This is particularly important for precipitation. Personally I am not sure if the data set is able to reproduce the elevational gradient of precipitation, which appears to be an important trigger of the lower treeline. I suggest showing precipitation and temperature maps and (if possible) comparing seasonal and annual data with available station data for a rough quality assessment.*

Rep.: We used the CHELSA dataset especially because it considers the elevational gradients and local relief. We think that it should be sufficient to mention the advantages using this dataset for our investigation in the manuscript. A detailed description of the used dataset can be found in the respective publication of the data: https://www.nature.com/articles/sdata2017122. This paper also included a wide variety of validation tests for the spatial and temporal performance of the used data compared to independent datasets. We do not have independent data available for Mongolia, so we cannot perform an independent comparison in this case. We also think that the dataset is sufficiently validated and peer reviewed and therefore would leave the validation for the reader to look up in the respective publication.

It would be no problem to provide any additional climate maps for the publication, maybe as supplement material. However, we would leave this decision to the editors.

We however attached a few figures to underline our selection of climate data:

[Figure]

Figure 1: Comparison of PRISM and CHELSA for Mongolia. PRISM is at a resolution of 4km and CHELSA at a resolution of 1km.

[Figure]

Figure 2: Small scale comparison of mean annual precipitation and Forest occurrence in the region of Ulanbatar. Visible are the dryer valleys and wetter mountain slopes. The lower Treeline in this region is at around 1650 m, what would correspond to a mean of 240mm of annual precipitation (Figure 3).

[Figure]

Figure 3: Annual precipitation vs. elevation in the region of Ulanbatar.

*Ref.: Further the generation of solar radiation data is not clear to me? Is it solely based on the DEM or is any atmospheric information included?*

Rep.: It was simple calculated from the DEM with a GIS-tool. We added this information to the text

*Ref.: 4) Ecozones and land cover classes: From my understanding, the larger scale "ecozones" are driven by climatic conditions (just like the land cover classes derived from remote sensing data). A habitat of a particular vegetation type (class) should actually occupy a certain climatic niche (regardless of the large scale "ecozone"). Please clarify why "ecozones" were used to separate the data.*

Rep.: You are right with the definition of the term "ecozones" and we changed the term all over the manuscript. At least we are focusing on the boreal forest ecosystems, so we change the term "ecozone" to "type of boreal forest" and the class "Total ecozone" to "Total ecosystem unit" throughout the entire manuscript.

*Ref.: Further, the results indicate different statistical relationships between treeline patterns and climate variables in different zones (inverted relationships, e.g. l.369ff). I wonder if there could be any ecological mechanism behind or whether this is rather a statistical artefact (especially since there is only one ecozone, which is mainly covered by forest).*

Rep.: There are different ecological mechanisms at different types of boreal forest. This is shown by different tree species composition and by different physical adaption of individual trees.

*Ref.: 5) Statistical Methods: The statistical methods for the analysis of climate-landcover relationships are parametrical, i.e. they require normally distributed input data (spatial data depend on the terrain and are certainly not normal distributed). This is relevant for the assessment of significant differences (Tab. 1) and also for the linear regressions (Fig. 4). I feel, the use of the regression is appropriate (since only the direction is discussed in the text), however, limitations should be clearly stated. Table 1 could be shown as boxplots and significance testing should be avoided.*

Rep.: As you can see in Figure 4 all of the data is mostly normally distributed. We added information about the restriction of the regression analysis.

*Ref.: 6) Spatial Interpolation of treeline elevation: The treeline is (as the analysis shows) highly influenced by local scale climate conditions. Thus the spatial interpolation seems to be misleading to me. I recommend illustrating the treelines as polygons.*

Rep: Spatial interpolation of the treeline as surfaces is standard method in Geography.

*Ref.: Further the potential anthropogenic influence on the treeline location should be elaborated.*

Rep.: We reported human impact on treeline and forest distribution as tree cutting, forest clearing, fire setting and wood pasture since long time.

*Ref.: Further in Table 3 spatial ratios of forest cover are calculated for the lower and upper treeline. I do not understand, what these ratios mean (the treeline is actually not an area, but an ecological border)?*

Rep: The constitution of boreal forest in Central Asia is strictly differentiated between more or less closed forest stands and grassland areas. Although the treeline may be visually a line on a small scale local view, depending on topographic effects, human impact and reproductive processes it covers an area of advancing and retreating forest. From the geobotanical view the fragmentation of forest stands is a main ecological factor for resilience in the ecotone. is As spatial boundary the treeline covers a distinct area where forests extend or reduce simultaneously over a short distance.

*Ref.: 7) Treelines and climate change: The authors use the spatial pattern of treelines and spatial variations of NDVI in order to discuss the potential impact of climate change on tree growth. For me this link is not trivial and should be better investigated. I recommend to also analyze the temporal variability of NDVI, e.g. with respect to warm, cold, dry or wet years during the observational period.*

Rep.:  Doing the analysis in the proposed detailed way is a method to be used at a more local scale and does not fit to our regional processing.

The spatial distribution of forest and treeline does not react to annual or seasonal extremes, they integrate a longer period. The NDVI of trees is furthermore strongly controlled by soil site conditions. This fact we have discussed in the manuscript. From field dendrochronological investigation we know that trees can suffer for several decades after one year of drought. How do indicate such an event exclusively by NDVI trend analysis?

On the one hand we had to generalize our data with respect to the broad investigation area; on the other hand more detailed analysis needs more precise data in spatial and temporal resolution, which does not exist. Thus our approach meets the most comprehensive method to delineate climate limitations at a regional scale. Moreover, modelling climate projections does not provide annual or seasonal data; it can only produce trends for periods. Therefore our "rough" data fits to the possibilities of these scenarios.

*Ref.: Further, the manuscript contains very little information about climate change scenarios (and potential changes of temperature and precipitation) for Central Asia. This would be an important basis for the discussion. Once again, please focus on results, which are really supported by your study! E.g. sentences in l. 491/492 or 498/499 (but also others) are very speculative and not proven by the analysis or by literature.*

Rep.: We added some information about climate change scenarios at those places, where it was necessary to substantiate our statements for potential developments in the future. However, it was not our aim to make any forecast into the future. We only want to point out that our results can be used by modelling future and former climate and environmental conditions by the distribution of boreal forest as proxydata. Our main aim was to focus on the recent forest under the climate, which can be seen in the field as well as in the remote sensing data, to show that the results of our investigation fit well to those from biologist and geomorphologist.

---

## Author Response (AR1)

**Authors response to the editor**

Dear Jochen Schöngart, thank very much for your support to manage our manuscript for potential publishing in the journal Biogeoscience. You can see in the document below, that we have changed a lot – even the title – in our manuscript since there have been many helpful comments from the reviewers. At least, we asked a native English-speaking colleague to check the spelling. We have already stated most of our corrections in the reply to the reviewers and you will find some of again listed below. However, after first revising line by line the issues mentioned by the 1st reviewer, we afterwards had to rearrange several parts in the text. This makes the marked changes in the document below quit confusing. Finally, the manuscript had gained from reworking.

Best regards,
Michael Klinge

General changes:

We have changed the term "ecozone" to "type of boreal forest" and the class "Total ecozone" to "Total ecosystem unit" throughout the entire manuscript.

We have rearranged many parts of the text to receive a more consistent structure. Repetitions and less relevant information were mostly deleted. We also tried to shorten the text. However, we had to insert more detailed description where our statements were unclear.

Figure 3 it is very important to give the reader a visual idea how the different dataset spatially fit to each other and how the treeline points and the forest boundary was mapped. We inserted more relations to Fig. 3 in the text where it was necessary.
Figure 6 was moved to supplementary material.
There is a new order of figures following the text structure.

We changed the Figures about climate data, which shall appear in the supplement material following your instructions.

We added some information about climate change scenarios at those places, where it was necessary to substantiate our statements for potential developments in the future.

Specific remarks to changes proposed by the reviewer 1:

*- line 54: ...strongly varies in space and time...*      -done

*- line 61-62: soil temperature, soil moisture and soil nutrients might also play a role*

    - We incorporated this fact with an additional sentence

*- line 64-65: usage of the term 'ecozone' is confusing. Regarding altitudinal zonation I suggest to use the term 'zone' or 'belt' (alpine zone or alpine belt), regarding horizontal zonation I suggest to use the term 'habitat' or 'zone' or another term since the term 'ecozone' is associated*

*with large-scale units (biomes) such as humid mid-latitudes, dry mid-latitudes etc.*

    - We changed the sentence and will now consequently use the terms "zone" for horizontal and "belt" for altitudinal zonation

*- line 67: altitudinal zones are not biomes, but zones or belts*  - changed

*- line 85: no comma after et al.*

– not changed, because this kind of formatting is demanded by BIOGEOSCIENCE and automatically processed by CITAVI

*- line 103: either no comma before which or comma after Spot VGT)* - changed

*- line 105: see line 103*                          - changed

*- line 128-130: language editing* – The sentence is not necessary and completely deleted

*- line 144: trends of instead of trends for*          - done

*- line 166: showed the NDVI to be well usable....*   - done

*- line 167: tree biomass of Mongolian forests*       - done

*- line 176: climatically restricted? I suggest to rewrite: ...'is delimited by a constellation of climatic threshold values'*                          - That is good, changed

*- line 177-178: reflect climate-ecological relationships and limitations* - changed

*- line 195: highly continental semi-humid*   - changed

*- line 196: with little snowfall*                - changed

*- line 205: ...are arranged in characteristic sequences along latitudinal and altitudinal gradients*

- changed

*- line 206: obovata kursiv*                      - changed

*- line 207: selectively? Please rewrite this sentence*  - changed to: .. locally as mountain taiga ..

*- line 211: intra-montane basins*               - changed

*- line 217: the terms playas and takirs should be explained*

– I preferred to delete the sentence, because this information is not really necessary

*- line 222: forest management*                  - changed

*- line 226/227: Please explain the increasing fire susceptibility*

- explained by climate warming, permafrost retreat, and insect calamities

*- line 241: Fig. 2*                          - not changed, because directly naming the figure

*- line 244: language editing*                   - changed and hopefully better described

*- line 259: In the upper elevational zones?*  - changed

*- line 265: tree species maps*                      - changed

*- line 268-270: Using this approach the authors should be aware of and should point out that this is a simplification since the plant species respond to inter-annual variations and extreme values; plant species do not respond to mean values*

- We mentioned this problem following your words; however, in a next step of research it would be interesting to investigate if it possible to detect single extreme values in the data, which may play a significant rule for limiting tree distribution. But first it needs to establish the multi-data analysis like shown here, before to go to deep into detailed analysis, while the accuracy of the base database does not fit the research problem.

*- line 293-294: language editing*               - changed, sentence shortened

*- line 298: multiple comparisons*              - changed

*- line 306-308: Alteration of treelines requires successful recruitment of tree species. The authors should be aware of the fact that bioclimatic requirements of seedlings and saplings might deviate to a considerable extent from those of adult trees*

- You are right. That this is true, we could see during our last fieldwork in Mongolia. We wanted to examine why there is such a bad rejuvenation for larch trees like observed 3 years ago. Now after three rainy and humid summers we found extensive succession and even larch seedlings inside the steppe. However, from the remote sensing point, we can only detect adult trees and forests, which must have had sufficient environmental conditions to survive for a longer period. At the end of this chapter method, we inserted a complete new paragraph, where we described the ecological problems of forest distribution, human impact, and the technical limits of the investigation presented here.

*- line 332: intermontane basins*          - changed

*- line 335: language editing*          - changed

*- line 340-341: language editing*          - changed

*- line 344: language editing*          - changed

*- line 347-348: language editing*          - changed

*- line 380: blank space*          - changed

*- line 381: forest distribution or forest stand distribution*     - changed

*- line 433ff: Ulmus trees along water courses in the steppes should also be mentioned here*

Ulmus trees play a minor role in our investigation, because they occur at water-favored places near river and in the basins. Therefore their occurrence is less climate depend and also the basin region were excluded from treeline analysis. However, we inserted the Ulmus trees in the introduction to the Study area.

*- line 447: intramontane basins*          - changed

*- line 472-473: language editing*          - changed

*- line 474: 2x thus*          - changed

*- line 474-475: hygrophilous instead of water-demanding*

            not changed because trees are not specific hygrophilous species

*- line 478: which additional factors?*

- We can only assume what the additional factor may be: permafrost, water from upper slope,… The sentence was changed to the meaning that we can identify the position but not the specific ecological exception of extraordinary forest stands.

*- line 479: results instead of tendencies*     - changed

*- line 496: language editing*          - changed

*- line 507: Climatic change will lead: : :.*     - changed

*- line 509: Forest dynamics*          - changed

*- line 510: modelled*          - changed

*- Fig. 5: Map legend: there is no reference to the black line (not all of the readers are familiar with the borders of Mongolia)*     - changed

*- Fig. 6: Legend: Pinus sibirica*          - changed

[revised manuscript text omitted]

---

## Author Response (AR2)

**Authors response to the editor**

Dear Jochen Schöngart,

thank you very much again for accepting our manuscript to be published in the journal Biogeoscience. I have corrected all the points you have listed. I hope that everything is now fitting well for the final publishing process. Please inform me, if there are still any problems.

Best regards,
Michael Klinge